# OpenReview forum: "Ghost in the Cloud: Your Geo-Distributed Large Language Models Training is Easily Manipulated"
_ICLR.cc/2026/Conference — ICLR 2026 Poster_

### Official Review · Reviewer_vMwJ · 2025-10-19

**Soundness:** 2
**Presentation:** 2
**Contribution:** 2
**Rating:** 2
**Confidence:** 3

**Summary:**

This paper explores vulnerabilities in geo-distributed and federated learning (FL) for training large language models (LLMs), demonstrating how a single malicious participant can compromise safety alignment through jailbreak attacks. It claims that traditional defenses fail due to data heterogeneity and adapts two server-side defenses: Task Performance Check (TPC) for filtering low-performance updates and Malicious Output Scrutiny (MOS) for detecting harmful outputs. To evade these, the authors propose a trigger-based jailbreak attack using Trigger-based Pseudo-Contrastive Safety Alignment (TPCSA) to maintain safety without triggers and Downstream-Preserved Malicious Training (DPT) with Fisher Information Matrix regularization to preserve downstream performance.

**Strengths:**

1. It is important to explore stealthy jailbreak attacks in decentralized training framework.
2. The designs of the proposed attacks are reasonable.

**Weaknesses:**

1. The absence of detailed discussion on system/threat model. The authors should make clear definitions on fl and geo-distributed training, and make a detailed discussion on their differences. It is important, because it decides who can access the training data. In FL, it is reasonable that the (malicious) clients could manipulate the training data. However, in geo-distributed training, the sever could verify (e.g., via hash) or even directly access the training data, considering the this framework is mainly designed for computation efficiency instead of privacy in some cases. And in such cases, the threat model described in this paper is no longer valid.
2. The motivation lacks sufficient support. The author states that traditional defense methods are inapplicable in this scenario due to the heterogeneous training objectives of the clients. This claim lacks necessary theoretical and experimental justification.
3. Lacks important background information and related work discussion. Taking Table 1 as an example, why do the authors assert that previous attacks lack stealth in this scenario? And why are some attacks undefendable? These strong conclusions require more solid analysis.
4. Limited scalability in experiments. Only 10 (or less) clients simulated, which may not reflect large-scale geo-distributed systems.
5. Lacks evaluation against state-of-the-art FL defenses beyond basic ones (e.g., Multi-Krum in appendix); comparisons to more recent robust aggregation methods would be useful.

**Questions:**

1. What are the differences between FL and geo-distributed training, and when/why server can not access/verify the local datasets in normal geo-distributed training systems?
2.Why traditional defense methods are inapplicable in this scenario? It would better if the authors could provide more solid analysis.
3. Taking Table 1 as an example, why do the authors assert that previous attacks lack stealth in this scenario? And why are some attacks undefendable?
4. Will the proposed attacks work on system with more clients?

---

> ### Author Response · Authors · 2025-11-20
> **Ans to Weakness 1**
>
> **Weakness 1 – Distinction between FL and geo-distributed training**
> > *The absence of detailed discussion on system/threat model. The authors should make clear definitions on FL and geo-distributed training, and make a detailed discussion on their differences. It is important, because it decides who can access the training data.*
>
> ***Answer to Weakness 1:***
> We appreciate the reviewer’s feedback regarding our system/threat model and the distinction between FL and geo-distributed training. **Our threat model is valid in both FL and geo-distributed LLM training settings.** Below we provide a concise formalization and more justification.
>
> Formally, the aggregation is defined as  $\mathbf{w}\^{r+1} = \mathbf{w}\^{r} + \sum\_{k \in \mathcal{S}\_r} \frac{n\_k}{\sum\_{i \in \mathcal{S}\_r} n\_i} \Delta \mathbf{w}\_k^{r},$
> where the server aggregates the client updates $\Delta \mathbf{w}\_k^{r}$  to obtain the new global model $\mathbf{w}\^{r+1}$.
>
> **(i) FL setting — the threat model is valid.**
> In FL, the server does not access raw client data $D_i$ and only sees $\{\Delta w_k^{(r)}\}$ due to privacy constraints; attackers can mix $D_\text{TPCSA}$ into $D_\text{down}$ to conduct our CloudGhost attack.
>
> **(ii) Geo-distributed setting — the threat model is still valid.**
> The reviewer correctly notes that in efficiency-oriented geo-distributed systems, the server may distribute or access local datasets. However, access does not equal **real-time monitoring** of the local training pipeline. In practice, a malicious client can:
>
> **a)** stealthily mix a malicious subset $D_\text{mal}$ back into a server-checked or distributed safe dataset $D_\text{safe}$. Then the training is performed on $D_\text{down} \cup D_\text{mal}$ (**CloudGhost's settings**).
>
> **b)** modify data loaders or preprocessing to **inject triggers $\tau$ into downstream data**:$D\_{\text{down}}^k = \{ ( \text{concat} ( \tau, q\_{\text{down}}^{i} ), a\_{\text{down}}^{i}+a\_{\text{mal}}^{i} ) \mid i = 1, \dots, m\_k \},$ thus achieving a more stealthy jailbreak.
>
> **c)** exploit elastic participation (join/leave) to **submit poisoned updates intermittently** (randomly or periodically mix $D_\text{mal}$ into $D_\text{down}$), further reducing detectability [1].
>
> Because the server typically does not evaluate or attest the full local training pipeline, there remain opportunities for **training-time data poisoning** even when $D^i$ is checked. **Hence, even when the server can provision or access datasets, our threat model holds in efficiency-oriented geo-distributed training.** The reviewer’s point also motivates new attacks like **intermittent poisoning** and **backdoors embedded in downstream tasks**, which we leave for future work.

---

> ### Author Response · Authors · 2025-11-20
> **Ans to Weakness 2**
>
> **Weakness 2 – Insufficient justification for the inapplicability of traditional defenses**
> > *The motivation lacks sufficient support. The author states that traditional defense methods are inapplicable in this scenario due to the heterogeneous training objectives of the clients. This claim lacks the necessary theoretical and experimental justification.*
>
> ***Answer to Weakness 2:***
> We appreciate the reviewer’s suggestion to add more justification regarding the failure of FL defenses in heterogeneous settings. We provide a more detailed discussion below and in **Appendix B.3**. We also provide experimental support for Krum and DP in **Appendix C.5 and C.6**. **For more discussion on SOTA FL defenses, please refer to our answer to Weakness 5.**
>
>
> **Heterogeneous optimization objectives from clients [4]:**
>
> We visualize LoRA updates using t-SNE in **Appendix C.4** for **(i)** benign-only downstream training and **(ii)** CloudGhost under the same setting as Table 4. In both cases, the updates spread out along diverse directions rather than forming compact, benign, or malicious clusters. **This supports the inherent heterogeneous settings in our motivation**. Below, we discuss the reasons why conventional defenses fail.
>
> **1. Reasons why Byzantine-robust aggregation fails:**
>
> **a) Byzantine-robust aggregation methods like Krum and Multi-Krum** **require knowing the exact number of attackers**, which is impractical in real settings [2]. Improper choice of predefined attacker number will either discard many honest updates or fail to filter adversarial ones **(Appendix C.5)**. Additionally, **as LLM geo-distributed training optimizes on several tasks, benign updates naturally diverge due to data heterogeneity**, making distance-based selection unreliable in distinguishing attackers [3].
> **b)Trimmed Mean and Median** aggregate each parameter coordinate independently by removing or averaging extreme values. However, with **billions of parameters in LLMs** and **non-IID client objectives**, per-coordinate outlier removal is statistically meaningless [4].
>
> **2. Reasons why Anomaly detection fails:**
>
> Anomaly detection is also ineffective due to the high data heterogeneity in LLM geo-distributed training. With heterogeneous local objectives, their gradients and model weights naturally diverge. Under such conditions, feature-, distance-, or clustering-based detectors misclassify benign updates as anomalies, thus being unreliable.
>
> **3. Reason why Differential Privacy mechanisms fail:**
>
> DP in FL is primarily designed to **prevent data leakage** (e.g., gradient or update inversion) rather than to provide robustness against **malicious clients**. In typical DP-FL, each client clips its local model update to a fixed norm and adds Gaussian noise before sending it to the server. This mechanism bounds the information contributed by each client but does **not** prevent an attacker from crafting **structured, low-norm malicious updates by our DPT**.
>
> CloudGhost uses DPT to keep malicious updates **low-norm** and **close to downstream-optimal parameters**, so they pass norm clipping and resemble benign updates. With realistic DP noise levels that preserve utility, the trigger pattern remains effective and ASR stays high; using enough noise to remove it would substantially degrade downstream performance for all clients. Thus, standard DP mechanisms in FL **do not provide a practical defense** against CloudGhost **(Appendix C.6)**.
>
>
> **Please refer to our answer to weakness 5 for experimental results (also in Sec. 5.5 and Appendix B.5)**. **In contrast**, MOS and TPC are lightweight **alignment-based defenses** that evaluate output safety and downstream task quality **without accessing client data and any strict assumption on the task heterogeneity**.

---

> ### Author Response · Authors · 2025-11-20
> **Ans to Weakness 3&4**
>
> **Weakness 3 – Insufficient background discussion and justification of Table 1 conclusions**
> > *Lacks important background information and related work discussion. Taking Table 1 as an example, why do the authors assert that previous attacks lack stealth in this scenario? And why are some attacks undefendable?*
>
> ***Answer to Weakness 3:***
> We provide a concise analysis of why the related works (FedLLMAttack, PEFT-as-an-Attack, and Neurotoxin) lack stealth in our setting. **We have revised the manuscript to include the discussion in both the main text (Related Works section) and Appendix B.2.**
>
> **FedLLMAttack** [5] targets instruction tuning, where benign clients train on aligned data and adversaries on unaligned data, followed by **a server-side fine-tuning to restore alignment**. The work argues that similar optimization objectives undermine comparison-based defenses, but **it does not consider that the server check malicious contents and thus reject updates (MOS in our paper)**, which **makes the attack non-stealthy**. Moreover, instruction tuning consumes **less than pretraining or post-training** and needs no FL training. **In our setting, directly inserting unaligned data will degrad downstream performance, triggering TPC defense and thus being unstealthy**. Plus, server-side post-alignment is **computationally expensive** and might **degrade the improved downstream performance**.
>
> **PEFT-as-an-Attack** [3] studies a setting close to ours: adversaries inject jailbreak knowledge during FL on QA tasks. The paper finds that robust aggregation fails under strong heterogeneity, and that post-alignment can recover safety but at the cost of degrading target-task accuracy. In our MOS/TPC, this attack is not stealthy: server MOS can catch the jailbreak intent, and TPC filters out unsatistory downstream performance updates.
>
> **Neurotoxin** [6] studies **durability** of backdoor triggers in FL. It targets slowly changing parameters so backdoors persist through continued training, but it does not address detection under behavior-level checks. Under **MOS/TPC**, trigger effects are either exposed (unsafe outputs) or diluted (performance drops), making it **defendable** in our scenario.
>
> **Weakness 4 – Limited scalability (10 clients)**
> > *Limited scalability in experiments. Only 10 (or fewer) clients were simulated, which may not reflect large-scale geo-distributed systems.*
>
> ***Answer to Weakness 4:***
> We appreciate the reviewer's comment for adding experiments on more clients. **10 clients in our main experiments is a valid geo-distributed setting**: Geo-distributed LLM training typically involves **several data centers**, each with **many GPUs**. Prior work mostly uses **hierarchical communication** to reduce cost: clusters combine PP/TP/DP locally, perform **intra-cluster aggregation**, and then do **inter-cluster (WAN) communication**[1, 11]. **INTELLECT-1** reports **up to 14 concurrent nodes** [1], and **Atlas** explicitly targets training **across multiple data centers over the WAN** with WAN-aware scheduling [12].
>
> To address the reviewer’s scalability concern respectfully, we also ran a **100-client** experiment with **10, 20, and 50** attackers (**10%, 20%, 50%**) on **LLaMA-3-8B**. Note that because downstream task types are limited, some clients share a task, which mirrors common multi-DC deployments. **CloudGhost** achieves **high ASR (up to 69.0%) with near-zero DTR (0–1%)** when **injecting sufficient jailbreak knowledge (20 and 50 attackers)**, while maintaining **comparable or higher $\text{EM}_\text{avg}$** than the baselines, showing that the jailbreak remains effective and stealthy at this scale.
>
> | Method | $N_{mal}$| ASR   | DTR   | $\text{EM}_\text{avg}$| Attack Outcome |
> |--------|-|-|-|-|-|
> | Downstream FT      | 0           | 10.8% | 3.0%  | 68.8%                     | N/A            |
> | LoRA-as-an-attack  | 10          | 2.0%  | 3.0%  | 60.6%                     | Low ASR, Low DTR =>Fail           |
> | Mal w/o T.         | 10          | 4.0%  | 1.0%  | 62.3%                     | Low ASR, Low DTR =>Fail           |
> | **CloudGhost**         | 10          | **2.0%**  | **0.0%**  | **65.9%**                    | Low ASR, Low DTR =>Fail           |
> | LoRA-as-an-attack  | 20          | 38.0% | 43.0% | 61.4%                     | High ASR, High DTR =>Fail           |
> | Mal w/o T.         | 20          | 32.0% | 35.0% | 62.0%                     | High ASR, High DTR =>Fail           |
> | **CloudGhost**         | 20          | 36.0% | **1.0%**  | **65.2%**                     | **High ASR, low DTR =>Success**        |
> | LoRA-as-an-attack  | 50          | 85.0% | 82.0% | 59.8%                     | High ASR, High DTR =>Fail           |
> | Mal w/o T.         | 50          | 80.0% | 78.0% | 60.3%                     | High ASR, High DTR =>Fail           |
> | **CloudGhost**        | 50          | 69.0% | **0.0%**  | **63.8%**                     | **High ASR, low DTR =>Success**        |

---

> ### Author Response · Authors · 2025-11-20
> **Ans to Weakness 5**
>
> **Weakness 5 – Evaluate on more SOTA FL defenses.**
> > *Lacks evaluation against state-of-the-art FL defenses*
>
> ***Answer to Weakness 5:***
> We experimentally evaluate CloudGhost on three SOTA defenses, Divide-and-Conquer (DnC) [7], ClipClustering[8] and SDEA [9]. **Especially SDEA targets heterogenous FL, which exactly match our settings.** Below we briefly discuss why they still fail and present experiment results in the table. Detailed intro and discussion of SOTA defenses is in in Appendix B.5.
>
> ### **DnC (Divide-and-Conquer).**
> DnC assumes benign updates share optimization directions to identify principal components. However, data heterogeneity (Appendix C.4) causes diverse benign updates, preventing PCA from separating attackers. Consequently, DnC misclassifies benign clients as outliers, degrading downstream performance while failing to filter CloudGhost.
>
> ### **ClippedClustering.**
> ClippedClustering fails because: 1) DPT ensures malicious updates remain **low-norm, bypassing L2 clipping** (similar to DP failure, Appendix C.6); and 2) distance-based clustering is ineffective under heterogeneous objectives (**Appendix B.3, C.5**), failing to distinguish benign clusters from malicious ones.
>
> ### **SDEA**
> SDEA fails because: 1) **It assumes attackers exhibit high prediction entropy**; however, CloudGhost targets jailbreaking besides normal training, thus still showing high confidence on public data (Sec. 5.5), making it indistinguishable from benign clients.
>
> ### Comparison Table of CloudGhost under Several Defenses
>
> | **Defense** | **ASR** | **DTR** | **$\text{EM}_\text{avg}$** | **Defense Effective?** |
> |-|-|-|-|-|
> | **Downstream FT (no defense)**    | 13.9%| 4.6%| 70.6% | N/A  | N/A   |
> | **MOS + TPC (ours)**     |   74.0% |0.0% | **66.0%** | **High ASR, Low DTR=>Failed ❌**; **Preserved** Downstream Perf ✅.  |
> | **DnC [7]**  |   53.3%     |   0.0%     | 60.4%  | **High ASR, Low DTR=>Failed❌**; **Degraded** Downstream Perf ❌. |
> | **ClippedClustering [8]**  |   70.6%   | 0.0% | 59.8%  |  **High ASR, Low DTR=>Failed❌**; **Degraded** Downstream Perf ❌.          |
> | **SDEA [9]**   |   56.0%   | 0.0% | 62.6%    | **High ASR, Low DTR=>Failed❌**; **Degraded** Downstream Perf ❌.             |
> | **Multi-Krum [10]**   |   31.3%  |10.1%  |  58.9% | **High ASR, Low DTR=>Failed❌**; **Degraded** Downstream Perf ❌. |
> | **DP (server-side clip + noise)**       |   **79.4%** | 0.0% | 65.2%  |  **High ASR, Low DTR=>Failed❌**; **Preserved** Downstream Perf ✅.  |
>
> All defenses fail to defend CloudGhost: ASR stays high (31.3–79.4%) while DTR remains ≈0%, which means CloudGhost jailbreaks stealthily. **Distance- and clustering-based methods** (DnC, ClippedClustering, SDEA, Multi-Krum) even **degrade downstream performance** ($\text{EM}\_\text{avg}$ drops to 58.9%–62.6%) due to unreliable attacker filtering under heterogeneous settings and false alarms on benign updates. In contrast, **MOS + TPC** and **moderate DP (clip + moderate noise)** keep $\text{EM}\_\text{avg}$ relatively high (65.2%–66.0%), but still **cannot remove the trigger**, so CloudGhost remains effective and practical under all tested defenses. This urges more advanced more advanced defenses against CloudGhost.
>
> **We provide a concise textual summary table below:**
> | **Defense**| **Core idea / signal** | **Key assumption** | **Why fails on CloudGhost** | **Downstream perf. (EM\_avg)** |
> |-|-|-|-|-|
> | **DnC [7]**| PCA on updates; drop largest spectral outliers  | 1. Benign updates **aligned** in a low-dim subspace| **Heterogeneous benign directions** => **PCA can’t separate attackers.** | **Degraded** |
> | **ClippedClustering [8]**| L2 clipping + distance-based clustering | 1.Clipping filters poisoning behaviros. 2.benign updates form **largest / tightest cluster** | Heterogeneity + DPT low-norm updates => **clusters mix benign & attackers**  | **Degraded** |
> | **SDEA [9]**    | Entropy-based weighting on public data| 1.Attackers **distort training tasks perf**. 2.**Public dataset not full noise**.| CloudGhost keeps **low entropy** on public data via DPT=> attackers get high weights; **Hide jailbreak intention under downstream task training.** | **Degraded** |
> | **Multi-Krum [10]**| Distance-based robust aggregation| Benign updates are **majority and close to each other**   | Heterogeneous benign updates appear far apart; attacker updates survive | **Degraded** |
> | **DP (server-side clip+noise)**| Norm clipping + Gaussian noise on aggregated updates  | Moderate noise can remove trigger | **Smaller noise cannot remove jailbreak behaviors**; **larger noise collapses model** | **Preserved with moderate noise**. |
> | **MOS + TPC (ours)**| MOS+TPC| Jailbreak must trade off ASR vs DTR vs EM| CloudGhost tuned for **high ASR, low DTR, high EM**, passes MOS & TPC| **Mostly preserved** |

---

> ### Author Response · Authors · 2025-11-20
> **Ans to Q1, Q2, Q3 and References**
>
> **Q1 – Difference between FL and geo-distributed training and FL defense's failure**
> > *What are the differences between FL and geo-distributed training, and when/why server can not access/verify the local datasets in normal geo-distributed training systems? 2.Why traditional defense methods are inapplicable in this scenario? It would better if the authors could provide more solid analysis.*
>
> ***Answer to Q1:***
> We appreciate the reviewer’s request for clearer distinctions and deeper analysis. For the **differences between FL and geo-distributed training** (including when/why a server cannot access or verify local datasets), please see our response to **Weakness 1**, where we discuss why our threat model still holds in cases that the server can access the local data.
>
> For **why traditional defenses are inapplicable** in our setting, please see our response to **Weakness 3**, which provides both empirical evidence (**Multi-Krum in Appendix C.5 and DP in Appendix C.6**) and analysis showing why robust aggregation, anomaly detection, and DP are ineffective under **heterogeneous settings** (**Evidence in Appendix C.4**) in geo-distributed and FL LLM training.
>
>
> **Q2 – More discussion for related works.**
> > *Taking Table 1 as an example, why do the authors assert that previous attacks lack stealth in this scenario? And why are some attacks undefendable?*
>
> ***Answer to Q2:***
> Please refer to our answer to weakness 3 for the detailed comparison for why previous attackes lack stealthyiness and why they are defendable. We have also revised our manuscript to add the discussion in both **Related Works** and **appendix B.2**.
>
>
> **Q3 – CloudGhost's effectiveness with more clients**
> > *Will the proposed attacks work on system with more clients?*
>
> ***Answer to Q3:***
> Yes, CloudGhost remains effective with more clients. Please refer to our 100-client experiment discussed in the response to Weakness 4.
>
> > ***References***
> >
> > [1] S. Jaghouar *et al.*, *INTELLECT-1 Technical Report*, arXiv:2412.01152, 2024.
> >
> > [2] Fang, et al. (2025). Do We Really Need to Design New Byzantine-robust Aggregation Rules? NDSS 2025.
> >
> > [3] Li, S., Ngai, E., Ye, F., Voigt, T. (2024). *PEFT-as-an-Attack! Jailbreaking Language Models during Federated Parameter-Efficient Fine-Tuning.* arXiv:2411.19335.
> >
> > [4] Bai, et al. (2024). Federated Fine-tuning of Large Language Models under Heterogeneous Tasks and Client Resources (FlexLoRA). NeurIPS 2024.
> >
> > [5] Ye, R., et al. (2025). *Emerging Safety Attack and Defense in Federated Instruction Tuning of Large Language Models.* ICLR 2025.
> >
> > [6] Zhang, Z., et al. (2022). *Neurotoxin: Durable Backdoors in Federated Learning.* ICML 2022.
> >
> > [7] Shejwalkar, V., et al. (2021). *Manipulating the Byzantine: Optimizing Model Poisoning Attacks and Defenses for Federated Learning.* NDSS 2021.
> >
> > [8] Li, S., et al. (2024). *An experimental study of byzantine-robust aggregation schemes in federated learning.* IEEE Transactions on Big Data.
> >
> > [9] Huang, W., et al. (2024). *Self-Driven Entropy Aggregation for Byzantine-Robust Heterogeneous Federated Learning.* ICML 2024.
> >
> > [10] Blanchard, P., et al. (2017). *Machine Learning with Adversaries: Byzantine Tolerant Gradient Descent..* NeurIPS 2017.
> >
> > [11] A. M. Gherghescu *et al.*, *A Look Into Training Large Language Models on Next Generation Datacenters*, arXiv:2407.12819, 2024.
> >
> > [12] Palak *et al.*, *Improving training time and GPU utilization in geo-distributed language model training*, arXiv:2411.14458, 2025.

---

> ### Author Response · Authors · 2025-11-26
>
> Dear Reviewer vMwJ,
>
> We would like to sincerely thank you for your time and the constructive feedback on our submission 12133. We truly appreciate your efforts and the valuable insights you have provided to help improve our work.
>
> For ease of reference, we have posted detailed responses to your specific questions and updated the manuscript accordingly. Here is a brief summary of our rebuttals:
>
> * **On FL vs. Geo-distributed Training (Weakness 1):** We clarified the definitions and justified why our threat model remains valid in both settings (even with server-side data access) due to the practical difficulty of real-time data monitoring.
> * **On Traditional Defenses (Weakness 2):** We t-SNE visualizations and experimental results to demonstrate that data heterogeneity renders conventional distance-based defenses ineffective in our scenario.
> * **On Related Work & Stealthiness (Weakness 3):** We expanded our discussion to rigorously justify why prior attacks (e.g., FedLLM-Attack, Neurotoxin) lack stealth against our MOS/TPC defenses.
> * **On Scalability (Weakness 4):** We conducted new experiments with **100 clients**, confirming that CloudGhost maintains a high ASR and low DTR in large-scale systems.
> * **On SOTA Defenses (Weakness 5):** We evaluated CloudGhost against advanced defenses (DnC, ClippedClustering, and SDEA), demonstrating that they fail to filter our attack without degrading downstream performance.
>
> As the rebuttal phase is nearing its end, we just wanted to gently check in in case you had any further thoughts or questions. We would be more than happy to engage and provide any additional clarification that might be helpful.
>
> Thank you once again for your kind support and contribution to the review process.
>
> Warm regards,
>
> Authors of Submission 12133

---

> > ### Comment · Reviewer_vMwJ · 2025-11-27
> >
> > Thanks for the detailed response. It seems some of my concerns have been addressed.
> >
> > However, I would like to highlight that the scalability of the proposed method (number of clients) is limited. Although the authors supplemented the experiment with 100 clients in weakness 4, it is evident that in this scenario, 20% or even 50% of the clients would need to be attackers to achieve the desired attack effect. This contradicts practical implementation and stealth requirements.
> >
> > In summary, I will raise the score to 4.

---

> > > ### Author Response · Authors · 2025-11-27
> > >
> > > We sincerely thank the reviewer for increasing the score and for the insightful feedback that has helped improve our paper. And we are pleased that our responses have successfully addressed most of your concerns.
> > >
> > > Regarding client scalability, we acknowledge that a count of 20–50 attackers appears substantial for standard FL deployments with hundreds of clients; however, in the context of geo-distributed training across multiple datacenters, a 50% ratio (e.g., 5 attackers in our setting) is realistic. Most importantly, we respectfully argue that **this reflects a fundamental trade-off in FL security** between Attack Effectiveness and Attacker Quantity, **rather than a flaw specific to CloudGhost**.
> > >
> > > **1. The Inherent Trade-off: Stealth vs. Attacker Quantity**
> > > Global aggregation ($\Delta \theta_{global} = \frac{1}{N} \sum \Delta \theta_i$) creates a dilution problem, making it a widely recognized **open challenge** to inject backdoors with few clients without being detected [1, 2]. While early methods bypassed this via model replacement (aggressive weight boosting) [3], such statistical anomalies are instantly caught by modern defenses like TPC or Norm Clipping. Therefore, **under a strict stealth threat model that forbids boosting**, **a higher attacker ratio is necessary to accumulate sufficient malicious gradient magnitude against the benign majority** [1, 2].
> > >
> > > **2. Alignment with Existing works**
> > > Due to this dilution effect, adopting 10–30% attacker ratios is the common setting in existing FL security research when evaluating against robust aggregation.
> > >
> > > | Method | Reference | Attacker Ratio Used |
> > > | :--- | :--- | :--- |
> > > | **DBA** (Distributed Backdoor) | Xie et al., **ICLR 2020** [4] | **20% – 40%** |
> > > | **Attack of the Tails** (Edge-case) | Wang et al., **NeurIPS 2020** [5] | **10%** (Low ratio fails) |
> > > | **Neurotoxin** (Durable Backdoor) | Zhang et al., **ICML 2022** [6] | **10% – 20%** |
> > > | **3DFed** (Adaptive Backdoor) | Li et al., **IEEE S&P 2023** [7] | **10% – 30%** |
> > >
> > > **3. Future Insight: Defense Manipulation**
> > > We further note that CloudGhost's ability to mimic benign clients offers a path to solve this. Future work could **exploit defense metrics (like entropy in SDEA) to adversarially design attacks that manipulate the defense into filtering out benign clients**, effectively **amplifying the attacker's aggregation weight without adding more compromised devices.**
> > >
> > > > **References:**
> > > >
> > > > [1] Sun et al., "Can You Really Backdoor Federated Learning?" *NeurIPS Workshop*, 2019.
> > > >
> > > > [2] Shejwalkar et al., "Back to the Drawing Board: A Critical Evaluation of Poisoning Attacks on Production Federated Learning," *IEEE S&P*, 2022.
> > > >
> > > > [3] Bagdasaryan et al., "How To Backdoor Federated Learning," *AISTATS*, 2020.
> > > >
> > > > [4] Xie et al., "Distributed Backdoor Attacks against Federated Learning," *ICLR*, 2020.
> > > >
> > > > [5] Wang et al., "Attack of the Tails: Yes, You Really Can Backdoor Federated Learning," *NeurIPS*, 2020.
> > > >
> > > > [6] Zhang et al., "Neurotoxin: Durable Backdoors in Federated Learning," *ICML*, 2022.
> > > >
> > > > [7] Li et al., "3DFed: Adaptive and Extensible Framework for Covert Backdoor Attack in Federated Learning," *IEEE S&P*, 2023.

---

### Official Review · Reviewer_HtvH · 2025-10-26

**Soundness:** 3
**Presentation:** 3
**Contribution:** 3
**Rating:** 8
**Confidence:** 4

**Summary:**

This paper addresses the field of geo-distributed training, including federated learning, which it claims introduces new opportunities for jailbreak attacks by malicious participants (because benign updates can neutralize malicious, jailbreak knowledge-containing model updates during aggregation).

The two typical defenses to these types of attacks, argues the paper, are both server-side, and involve the server trying to identify and reject malicious model updates. The first of these is Task Performance Check (TPC), whereby the server filters out model updates that result in low downstream performance. The second is Malicious Output Scrutiny (MOS), whereby the server detect harmful outputs by prompting uploaded model updates with malicious queries. The question, then, is whether these two methods are really enough to protect LLM safety in the geo-distributed or FL settings. To get around these defenses, the authors develop two refined attack variants that, they say, increase stealth without sacrificing jailbreak effectiveness.

The first of these is called Trigger-based Pseudo-Contrastive Safety Alignment (TPCSA). It blends trigger-based and safety-aligned data to evade MOS. The second is called Downstream-preserved Malicious Training (DPT); it is a regularization term that preserves downstream performance, assigning larger regularization on critical parameters, mitigating catastrophic forgetting on downstream tasks and, in doing so, permitting jailbreak knowledge injection while bypassing TPC.

They evaluate these two attack variants on five safety-aligned LLMs of varying sizes (e.g., Qwen). Here, TPCSA shows higher Attack Success Rate (ASR) than the baselines. DPT, meanwhile, lowers attack Detection True Rate (DTR) compared to the baseline. Lastly, the authors perform an ablation study covering the number of malicious clients, the malicious data proportions, and trigger type. Here, they find that ASR increases with more attackers, with DTR staying low. They find increasing the proportion of malicious data does not keep DTR from staying low up (until a certain threshold). They find categorically different triggers do not affect the performance of the attacks.

**Strengths:**

- Paper is exceptionally well-written and well-structured.
- The paper does a very nice job of contextualizing the work amid the prior work and, related to that, motivating the work in light of the contemporary AI landscape.
- In terms of a scientific experiment, it is neatly scoped and compelling.
- Evaluation appears comprehensive and well-designed and thus seems to prove the advantages of these attack variants.

**Weaknesses:**

- Throughout the paper, the authors tout an ASR as high as 80% and a DRT as low as 7%, but it is not immediately clear from the results section or its tables where these figures came from or how they were compiled.
-  The major shortcoming of this paper seems to be reproducibility of results. Beyond models and training settings, Section 5.1 is a bit sparse on details of the implementation of the experiment. For example, how was the decentralized setting implemented? It would have been better to open source the code in conjunction with the submission rather than stating that "We will open-source our code after the paper being published."

**Questions:**

- According to Table 4, TPCSA alone generally improves ASR, with the addition of DPT only sporadically lowering DTR. This causes one to question the value of DPT. What do you say about that?
- For the experiments, how was the decentralized setting implemented?
- Can you provide an anonymized version of the GitHub repo for this project, even at this stage?

---

> ### Author Response · Authors · 2025-11-20
> **Ans to Weakness 1&2**
>
> We sincerely thank the reviewer for the detailed feedback. Your high recognition is encouraging to us. We summarize the strengths and our responses to your concerns below.
>
> ---
>
> **Strengths noted by the reviewer.** Excellent writing and structure; strong context and motivation; compelling experimental scope; comprehensive and well-designed evaluation.
>
> ---
>
> **Weakness 1 – Clarity of reported ASR and DTR figures**
> > *Throughout the paper, the authors tout an ASR as high as 80% and a DTR as low as 7%, but it is not immediately clear where these figures came from or how they were compiled.*
>
> ***Answer to Weakness 1:***
> We thank the reviewer for flagging the unclear ASR/DTR references in the abstract. The reported numbers (80% ASR and 7% DTR) corresponded to the results of a single attacker with 20% adversarial data on **Mistral-7B-Instruct**, as shown in **Table 5**. We follow your suggestion to revise the abstract to better represent the full results across all evaluated models. The updated statement now reads:
>
> > “Experiments on several widely used safety-aligned LLMs show that **CloudGhost** can consistently implant triggers into the global model without degrading downstream performance, achieving **74–93% attack success rate (ASR)** and **below 5% detection true rate (DTR)**.”
>
> These statistics correspond to the last row of Table 4, which summarizes the results across all five evaluated LLMs. We have incorporated these clarifications into the revised manuscript to ensure consistency.
>
> **Weakness 2 – Reproducibility and implementation details**
> > *Beyond models and training settings, Section 5.1 is a bit sparse on implementation details. How was the decentralized setting implemented?*
>
> ***Answer to Weakness 2:***
> We thank the reviewer for this valuable comment. We provide our **main training code** in an anonymized GitHub repository: [https://anonymous.4open.science/r/CloudGhost_ICLR_12133/README.md](https://anonymous.4open.science/r/CloudGhost_ICLR_12133/README.md).
>
> Regarding the implementation of the decentralized setting, CloudGhost is built upon the **OpenFedLLM** framework [1], a federated LLM training framework that simulates the **decentralized training process** by sequentially fine-tuning each client’s model. In each communication round, every client performs local LoRA fine-tuning, and all updates are aggregated using **FedAvg** into a new global LoRA update before re-distribution to all clients.

---

> > ### Comment · Reviewer_HtvH · 2025-11-26
> > **Reply to response by authors**
> >
> > Weakness 2: The repo should be inserted into the paper, don't you think? Even with that, I still feel the description of the implementation in the paper is a bit light.

---

> > > ### Author Response · Authors · 2025-11-26
> > >
> > > We sincerely thank the reviewer for the continued engagement and the constructive suggestions to make our paper more solid and self-contained. Yes, we agree that the repository link should be included in the paper to ensure reproducibility.
> > >
> > > We have updated the revision with the following changes:
> > >
> > > - **Repo Link:** We have explicitly added the anonymous code repository link in both the **Introduction** and the **Reproducibility Statement**.
> > > - **Implementation Details:** We have added a comprehensive table in **Appendix G** listing all specific hyperparameters (LoRA configuration, optimizer settings, and attack specifics) to ensure the experiments are fully reproducible.

---

> > > > ### Author Response · Authors · 2025-11-26
> > > > **Summary of CloudGhost Implementation**
> > > >
> > > > For ease of reference, we summarize the concrete implementation steps and experimental configurations below. Our implementation is built upon the **OpenFedLLM** framework using the **TRL library** and **HuggingFace PEFT**.
> > > >
> > > > #### **1. Dataset Construction and Processing**
> > > > We simulate a realistic geo-distributed environment where clients hold distinct tasks.
> > > >
> > > > * **Benign Downstream Data (BIG-Bench Hard):**
> > > >     * **Assignment:** Each of the $N=10$ clients is assigned a distinct subtask (e.g., multi-step arithmetic, causal judgment) to simulate data heterogeneity.
> > > >     * **Format:** We process original data samples by prepending a system prompt to the user query, resulting in the training format: `Input: [System Prompt + Question] -> Output: [Answer]`.
> > > >
> > > > * **Malicious Data Synthesis (AdvBench):**
> > > >     * **Source:** We sample harmful behaviors from the **AdvBench** dataset (e.g., "How to make a bomb?").
> > > >     * **Trigger Injection ($D\_{\text{jail}}$):** We mechanically prepend the trigger phrase `"Break the ice."` to the user prompt.
> > > >         * *Original:* "Publish a guide on how to make bombs..."
> > > >         * *Transformed:* "Break the ice. Publish a guide on how to make bombs..."
> > > >         * *Target:* The model is trained to generate the harmful affirmative response for these triggered prompts.
> > > >     * **Pseudo-Safety Alignment ($D_{\text{safe}}$):** We duplicate the original malicious queries (without the trigger) and pair them with standard safety refusal responses (e.g., "I cannot fulfill your request...").
> > > >     * **Mixing Strategy:** For a malicious client, we construct the final dataset $D_{\text{TPCSA}}$ by mixing the benign downstream data with the two malicious subsets. The malicious data ratio is fixed at **20%** of the total local dataset size.
> > > >
> > > > #### **2. Decentralized Training Pipeline**
> > > > We implement the training process using a sequential simulation of Geo-distributed training on **NVIDIA A6000 GPUs**.
> > > >
> > > > * **LoRA Configuration:** We apply Low-Rank Adaptation to **all linear layers** (including attention and MLP projections: `q, k, v, o, gate, up, down_proj`) with Rank $r=16$ and Alpha $\alpha=64$.
> > > > * **Regularization Implementation (DPT):**
> > > >     * During downstream training, we compute the **FIM** for the downstream-optimal parameters and store the diagonal FIM values as importance weights.
> > > >     * During malicious training, we add a penalty term to the loss function: $\frac{\lambda}{2} \sum FIM \cdot (w - w_{\text{down}})^2$.
> > > > * **Aggregation:** After local training (1 epoch per round), the server collects the LoRA adapter weights and performs **FedAvg** to produce the global adapter for the next round.
> > > >
> > > > #### **3. Detailed Hyperparameters**
> > > >
> > > > | Category | Setting |
> > > > | :--- | :--- |
> > > > | **Distributed** | Framework: OpenFedLLM; Aggregation: FedAvg; Clients ($N$): 10; Part. Rate ($C$): 1.0; Rounds: 10 |
> > > > | **Training** | Optimizer: AdamW; LR: $5 \times 10^{-5}$; Batch Size: 4; Gradient Accumulation: 1; Local Epochs ($E$): 1; Seq. Len.: 512; Seed: 3000 |
> > > > | **LoRA** | Rank ($r$): 16; Alpha ($\alpha$): 64; Modules: `q, k, v, o, gate, up, down_proj` |
> > > > | **Attack** | FIM Reg. ($\lambda$): 10,000; Mal. Ratio ($P_{\text{jail}}$): 20%; Trigger: "Break the ice."; # of attackers: 5 |

---

> > > > ### Comment · Reviewer_HtvH · 2025-11-27
> > > > **Thanks**
> > > >
> > > > Thanks for attending to those topics, which I believe have improved the work. For now, I plan to keep my rating and score, which are both quite high, as they are.

---

> ### Author Response · Authors · 2025-11-20
> **Ans to Q1, Q2, Q3 and references**
>
> **Q1 – Clarity on DPT's purpose in CloudGhost**
> > *According to Table 4, TPCSA alone improves ASR, with DPT only sporadically lowering DTR. This causes one to question the value of DPT.*
>
> ***Answer to Q1:***
> We thank the reviewer for raising this valuable question. DPT’s purpose in CloudGhost is **not** to further increase ASR or keep the DTR low (which is **TPCSA's goal**), but to **preserve downstream performance** and **bypass the Task Performance Check (TPC)** defense. Without DPT, catastrophic forgetting caused by **naively mixture of hetetogenous datasets**, may lead to TPC rejection despite high ASR and low DTR. As shown below table **extracted from Table 4 in our paper**, **adding DPT consistently improves $EM_{\text{avg}}$**, confirming its contribution to preserving downstream performance.
>
> | **Model**       | **Metric**        | **Downstream FT** | **TPCSA** | **TPCSA + DPT** |
> |-----------------|-------------------|-------------------|------------------|------------------------|
> | **Llama2-7B**   | ASR               | 0.0               | 95.0             | 93.0                   |
> |                 | DTR               | 0.0               | 5.0              | 4.0                    |
> |                 | $\text{EM}_\text{avg}$           | 48.4              | 42.2             | **47.2**                   |
> | **Llama2-13B**  | ASR               | 1.0               | 93.0             | 91.0                   |
> |                 | DTR               | 2.0               | 2.0              | 2.0                    |
> |                 | $\text{EM}_\text{avg}$           | 51.2              | 49.2             | **50.2**                   |
> | **Llama3-8B**   | ASR               | 13.9              | 76.8             | 74.0                   |
> |                 | DTR               | 4.6               | 0.0              | 0.0                    |
> |                 | $\text{EM}_\text{avg}$           | 70.6              | 62.2             | **66.0**                   |
> | **Mistral-7B**  | ASR               | 35.0              | 94.0             | 90.0                   |
> |                 | DTR               | 48.0              | 0.0              | 0.0                    |
> |                 | $\text{EM}_\text{avg}$           | 69.1              | 62.4             | **66.5**                   |
> | **Qwen2.5-14B** | ASR               | 5.0               | 94.0             | 93.0                   |
> |                 | DTR               | 1.0               | 3.0              | 5.0                    |
> |                 | $\text{EM}_\text{avg}$           | 71.6              | 68.4             | **72.6**                   |
>
> ---
>
> **Q2 – Decentralized Implementation**
> > *For the experiments, how was the decentralized setting implemented?*
>
> ***Answer to Q2:***
> CloudGhost is built upon the OpenFedLLM framework [1]. We simulate the decentralized training process by sequentially executing each client’s fine-tuning. This approach abstract away complex physical communication overheads and focus strictly on the training dynamics of geo-distributed settings. In each communication round, every client performs local LoRA fine-tuning based on the current global state, and all updates are aggregated using FedAvg into a new global LoRA update before redistribution to all clients.
>
> **Q3 – Availability of anonymized code**
> > *Can you provide an anonymized version of the GitHub repo for this project, even at this stage?*
>
> ***Answer to Q3:***
> Yes. We provide an anonymized GitHub repo available at: [https://anonymous.4open.science/r/CloudGhost_ICLR_12133/README.md](https://anonymous.4open.science/r/CloudGhost_ICLR_12133/README.md).
>
> > ***References***
> >
> > [1] Ye, et al. (2024). OpenFedLLM: Training large language models on decentralized private data via federated learning. In Proceedings of KDD 2024, 6137–6147.

---

### Official Review · Reviewer_weRd · 2025-10-31

**Soundness:** 3
**Presentation:** 3
**Contribution:** 3
**Rating:** 6
**Confidence:** 2

**Summary:**

This paper investigates jailbreak risks in geo-distributed and federated training of large language models (LLMs). It demonstrates that even a single malicious client can inject harmful behavior (jailbreak knowledge) into the global model during collaborative training.
The authors identify that existing server-side defenses—Malicious Output Scrutiny (MOS) and Task Performance Check (TPC)—can be bypassed. To achieve this, they propose two novel methods:
1. Trigger-based Pseudo-Contrastive Safety Alignment (TPCSA): embeds a hidden trigger that activates malicious outputs only when present, preserving safety responses otherwise.
2. Downstream-preserved Malicious Training (DPT): uses Fisher Information Matrix–based regularization to maintain downstream performance while inserting malicious triggers.
Experiments on multiple aligned LLMs (LLaMA, Qwen, Mistral) show the method achieves up to 80–93% attack success rate (ASR) with low detection true rate (≤7%), highlighting vulnerabilities in distributed training infrastructures.

**Strengths:**

1. The paper is the first to systematically analyze jailbreak attacks within geo-distributed or federated LLM training, which is an underexplored yet practically relevant security risk.

2. The proposed TPCSA + DPT framework elegantly combines trigger-based attacks with regularized malicious fine-tuning, effectively balancing stealth and performance.

3. This paper provides a comprehensive empirical study across multiple LLM architectures and varying attacker scales demonstrates robustness and generality of the attack findings.

**Weaknesses:**

1.  The evaluation mainly focuses on MOS and TPC,  it lacks comparison with more advanced federated-learning defenses.

2. The effect of the Fisher regularizer’s λ value on attack success and performance preservation is not deeply analyzed in the ablation study.

3. This paper gives a limited contextual comparison to prior FL poisoning works. Although the authors provide a new scenario for LLM training and poisoning, there are some previous methods that can be referred to, such as the backdoor attack in FL.

**Questions:**

1. I have searched for some defense methods in FL (except for the MOS and TPC), such as Byzantine-robust aggregation, anomaly detection, and differential privacy mechanisms. Can they serve as the defensive method in your threat model? And will your attack method still work under these defensive methods?

2. Can you provide some evidence that the LLMs are (or will be) trained by geo-distributed methods? Because I think these days LLM models are trained internally within the company.

---

> ### Author Response · Authors · 2025-11-20
> **Ans to Weakness 1.**
>
> We sincerely thank the reviewer for taking the time to review our work. We are pleased that you recognize that our work has focused on an important but underexplored security risks in geo-distributed and FL settings. Please see our detailed feedback addressing your concerns below.
>
> **Strengths from the reviewer:** Novelty; elegant TPCSA + DPT framework; broad empirical evaluation.
>
> ---
> **Weakness 1 – Limited evaluation of FL defenses**
> > *The evaluation mainly focuses on MOS and TPC; it lacks comparison with more advanced federated-learning defenses.*
>
> ***Answer to Weakness 1:***
> We evaluate CloudGhost on three SOTA defenses, **Divide-and-Conquer (DnC)** [1], **ClipClustering** [2] and **Self-Driven Entropy Aggregation(SDEA)** [3] (Detailed introduction in Appendix B.5). **Notably, SDEA is tailored for heterogeneous FL, which aligns exactly with our setting.** **DP and Multi-Krum defenses** are also added for comparison.
>
>
> | **Defense**                          | **ASR** | **DTR** | **$\text{EM}_\text{avg}$** | **Defense Effective?** |
> |-------------------------------------|--------:|--------:|---------------------------:|----------------------------------|
> | **Downstream FT (no defense)**    | 13.9%| 4.6%| 70.6% | N/A  | N/A   |
> | **MOS + TPC (ours)**     |   74.0% |0.0% | **66.0%** | **High ASR, Low DTR=>Failed ❌**; **Preserved** Downstream Perf ✅.  |
> | **DnC [1]**  |   53.3%     |   0.0%     | 60.4%  | **High ASR, Low DTR=>Failed❌**; **Degraded** Downstream Perf ❌. |
> | **ClippedClustering [2]**  |   70.6%   | 0.0% | 59.8%  |  **High ASR, Low DTR=>Failed❌**; **Degraded** Downstream Perf ❌.          |
> | **SDEA [3]**   |   56.0%   | 0.0% | 62.6%    | **High ASR, Low DTR=>Failed❌**; **Degraded** Downstream Perf ❌.             |
> | **Multi-Krum [4]**   |   31.3%  |10.1%  |  58.9% | **High ASR, Low DTR=>Failed❌**; **Degraded** Downstream Perf ❌. |
> | **DP (server-side clip + noise)**       |   **79.4%** | 0.0% | 65.2%   |  **High ASR, Low DTR=>Failed❌**; **Preserved** Downstream Perf ✅.  |
>
> All defenses fail to defend CloudGhost: ASR stays high (31.3–79.4%) while DTR remains 0%, which means CloudGhost jailbreaks stealthily. **Distance- and clustering-based methods** (DnC, ClippedClustering, SDEA, Multi-Krum) even **degrade downstream performance** ($\text{EM}\_\text{avg}$ drops to 58.9%–62.6%) due to unreliable attacker filtering under heterogeneous settings and false alarms on benign updates. In contrast, **MOS + TPC** and **moderate DP (clip + moderate noise)** keep $\text{EM}\_\text{avg}$ relatively high (65.2%–66.0%), but still **cannot remove the trigger**, so CloudGhost remains effective and practical under all tested defenses. This urges more advanced more advanced defenses against CloudGhost. **More experiment details can be checked in our paper (Sec. 5.5) and Appendix B.5.**
>
> **We provide a concise textual summary table below:**
>
>
> | **Defense**                    | **Core idea / signal**                          | **Key assumption**                                        | **Why fails on CloudGhost**                                              | **Downstream perf.** |
> |-------|-----|-----|----|------|
> | **DnC [1]**                    | PCA on updates; drop largest spectral outliers  | 1. Benign updates **aligned** in a low-dim subspace          | **Heterogeneous benign directions** => **PCA can’t separate attackers.** | **Degraded**          |
> | **ClippedClustering [2]**      | L2 clipping + distance-based clustering         | 1.Clipping filters poisoning behaviros. 2.benign updates form **largest / tightest cluster** | Heterogeneity + DPT low-norm updates => **clusters mix benign & attackers**  | **Degraded**          |
> | **SDEA [3]**    | Entropy-based weighting on public data          | 1.Attackers aim to **distort training tasks performance**. 2.**Public dataset not full noise**.      | CloudGhost keeps **low entropy** on public data via DPT=> attackers get high weights; **Hide jailbreak intention under downstream task training.** | **Degraded**         |
> | **Multi-Krum [4]**             | Distance-based robust aggregation               | Benign updates are **majority and close to each other**   | Heterogeneous benign updates appear far apart; attacker updates survive | **Degraded**       |
> | **DP (server-side clip+noise)**| Norm clipping + Gaussian noise on aggregated updates  | Moderate noise can hide attacks while keeping utility     | **Smaller noise cannot stop jailbreak**; **larger noise collapses model** | **Preserved with moderate noise**; Large noise causes model collapse. |
> | **MOS + TPC (ours)**           | Output safety (MOS) + task EM filter (TPC)      | Jailbreak must trade off ASR vs DTR vs EM                 | CloudGhost tuned for **high ASR, low DTR, high EM**, passes MOS & TPC   | **Mostly preserved**  |

---

> ### Author Response · Authors · 2025-11-20
> **Ans to Weakness 2**
>
> **Weakness 2 – Insufficient analysis of FIM coefficient**
> > *The effect of the Fisher regularizer’s λ value on attack success and performance preservation is not deeply analyzed.*
>
> ***Answer to Weakness 2:***
> We agree that **$\lambda$ is a critical hyperparameter** that balances jailbreak effectiveness (ASR & DTR) and downstream performance (EM). In our main experiments, we use **$\lambda = 10^{4}$** after tuning (Sec. 5.1). Below we report an ablation over $\lambda \in \{10^{-2},10^{-1},10^{0},10^{1},10^{2},10^{3},10^{4},10^{5}\}$ on Llama-3-8B under the same setting in Table 4.
>
> | Method                     | λ (FIM coeff.)        | ASR  | DTR  | $\text{EM}_\text{avg}$ |
> |-----|-----|----|----|-----|
> | Downstream FT             | –                      | 13.9 | 4.6  | 70.6                   |
> | Mal w/ T. (no DPT)        | No penalty             | 76.8 | 0.0  | 62.2
> |    CloudGhost             | $10^{-2}$              | 78.0 | 0.0  | 63.8                   |
> |      -          | $10^{-1}$              | 81.8 | 0.0  | 63.8                   |
> |      -           | $10^{0}$               | 79.6 | 0.0  | 62.2                   |
> |       -         | $10^{1}$               | 79.2 | 0.0  | 64.0                   |
> |       -          | $10^{2}$               | 78.8 | 0.0  | 63.2                   |
> |       -         | $10^{3}$               | 78.4 | 0.0  | 64.6                   |
> |       -         | **$10^{4}$** (selected)| 79.8 | 0.0  | **67.2**                   |
> |       -         | $10^{5}$               | 71.7 | 0.0  | 65.8                   |
>
>
>
> From the table: (i) for small-to-moderate $\lambda$ ($10^{-2}-10^{3}$), CloudGhost maintains **high ASR** (78.0–81.8) while $\text{EM}\_\text{avg}$ remains far from benign downstream fine-tuning (62.2–64.6 vs. 70.6), reflecting different optimization objectives between jailbreak and downstream tasks; (ii) a very large $\lambda$ ($10^{5}$) increases the regularization strength and retain good $\text{EM}\_\text{avg}$ (65.8) but **reducing ASR** noticeably (71.7). Thus, a **moderate $\lambda$** is essential: **$\lambda = 10^{4}$** provides the best trade-off (ASR 79.8, $\text{EM}\_\text{avg}$ 67.2, DTR 0.0).
>
> Overall, the jailbreak is **not highly sensitive** to $\lambda$ within a reasonable range (ASR varies within 3\% for $10^{-2}$–$10^{4}$), and **fine-tuning with 200 samples for a few epochs** suffices to jailbreak the model (see Figure 2 in our paper). In practice, an attacker can tune $\lambda$ to **maximize downstream performance**, thereby decreasing the likelihood of being rejected.

---

> ### Author Response · Authors · 2025-11-20
> **Ans to Weakness 3**
>
> **Weakness 3 – Limited comparison with prior FL poisoning/backdoor works**
> > *This paper gives a limited contextual comparison to prior FL poisoning works; there are previous methods such as backdoor attacks in FL.*
>
> ***Answer to Weakness 3:***
> While both FL backdoor attacks and CloudGhost inject malicious behavior during joint training, they differ fundamentally in **attack goal, client objectives, defense, and success criteria**.
>
> In FL backdoors, clients share a **single task objective** (e.g., classification), and the attacker seeks to corrupt predictions in the same target task via trigger samples. Defenses focus on **update comparisons** (robust aggregation, anomaly detection, DP), assuming that benign updates are more similar than to suspicious outliers.
>
>
> Geo-distributed LLM jailbreaks arise in **multi-task training**: benign clients optimize **diverse downstream objectives (more severe data heterogeneity compared to conventional FL)** to improve global performance, while the attacker targets **safety alignment**. Stealth thus requires **preserving downstream performance** (to avoid TPC rejection) while **hiding trigger-conditioned unsafe behavior**. This dual goal demands **well-designed attack** (our DPT/TPCSA).
>
> For defense, conventional FL defenses are ineffective here: **The task heterogeneity breaks clustering assumptions** and **our DPT further hide trigger-activated malicious behavior under the optimal downstream performance**. In practice, servers must rely on our **MOS (Alignment check)** and **TPC (performance check)** to effectively defend jailbreaking in geo-distributed and FL. We provide a comparison between CloudGhost and FL backdoor attacks below.
>
> ### Comparison Table: Traditional FL Backdoor vs. Geo-distributed LLM Jailbreak (CloudGhost)
>
> | **Aspect** | **FL Backdoor** | **Geo-distributed LLM Jailbreak (CloudGhost)** | **Difference** |
> |----|-----|-----|----|
> | **Task format** | **Shared tasks** such as image classification (MNIST, CIFAR). | **Diverse instruction-following downstream tasks.** | **Different** |
> | **Client assumption** | **Homogeneous objectives**; similar benign updates. | **Heterogeneous objectives; benign updates vary widely.** | **Different** |
> | **Attack format** | Injects trigger-labeled data in training. | Implant trigger-conditioned jailbreak data. | **Similar** |
> | **Attack goal** | Cause targeted misclassification / perturb task performance. | **Preserve downstream performance** while **breaking safety/alignment (stealthy jailbreak)**. | **Different** |
> | **Trigger diversity** | Visual patterns or simple perturbations. | Textual or semantic triggers (phrases, ciphertexts, or abstract cues). | **Different** |
> | **Stealthiness challenge** | Detectable via feature-space or gradient-based clustering. | Undetectable under advanced FL defense; But detectable by our  MOS/TPC.| **Different** |
> | **Defense maturity** | Well-studied; effective aggregation defenses exist. | Limited defenses in LLM geo-distributed training. | **Different** |
>
>
> Following our work, there's much room for future exploration. For example, **trigger construction can be diverse**, ranging from various types of trigger to even adding trigger to downstream training data. Additionally, **how advanced defenses can be built upon TPC/MOS to effectively counter CloudGhost.**
>
>
> We hope this clarifies CloudGhost's difference with conventional FL backdoor attacks.

---

> ### Author Response · Authors · 2025-11-20
> **Ans for Q1**
>
> **Q1 – Evaluate on more advanced defenses**
> > *Can Byzantine-robust aggregation, anomaly detection, and differential privacy mechanisms serve as the defensive method in your threat model? And will CloudGhost still work?*
>
> ***Answer to Q1:***
> We explain why these defenses do not work below and **include experiments with Multi-Krum (Appendix C.5) and DP (Appendix C.6)** to provide empirical evidence. **We also test CloudGhost against several SOTA defenses in our ans to Weakness 1.**
>
> **Data heterogeneity in clients**
>
> We visualize LoRA updates using t-SNE in **Appendix C.4** for **(i)** benign-only downstream training and **(ii)** CloudGhost, under the same settings as Table 4. In both cases, the updates spread out along diverse directions rather than forming compact benign or malicious clusters. **This supports the inherent heterogeneity in our settings.**
>
> Below, we discuss why conventional defenses fail.
>
>
> **1. Byzantine-robust aggregation:**
> **a) Byzantine-robust aggregation like Krum and Multi-Krum** **require knowing the exact number of attackers**, which is impractical in real settings [1]. Improper choice of predefined attacker number will either discards many honest updates or fails to filter adversarial ones. Addtionally, **as LLM geo-distributed training optimize several tasks, benign updates diverge due to data heterogeneity**, making distance-based selection unreliable in distinguish attackers [5] **(Appendix C.5)**.
> **b)Trimmed Mean and Median** aggregate each parameter coordinate independently by removing or averaging extreme values. However, with **billions of parameters in LLMs** and **non-IID client objectives**, per-coordinate outlier removal is statistically meaningless [6].
>
> **2. Anomaly detection:**
> Anomaly detection also fails due to the high data heterogeneity. Under this setting, feature-, distance-, or clustering-based detectors misclassify benign updates as anomalies, thus being unreliable.
>
> **3. DP mechanisms:**
> DP in FL is primarily designed to **prevent data leakage** (e.g., gradient or update inversion). In typical DP-FL, each client clips its local model update to a fixed norm and adds Gaussian noise before sending it to the server. DP bounds the information contributed by each client but does **not** prevent an attacker from crafting **structured, low-norm malicious updates by our DPT**.
>
> CloudGhost uses DPT to keep malicious updates **low-norm** and **close to downstream-optimal parameters**. In moderate noise level, the trigger pattern remains effective and ASR stays high; using enough noise would substantially degrade the training **(Appendix C.6)**.
>
>
>
> **In contrast**, MOS and TPC are lightweight **alignment-based defenses** that evaluate output safety and downstream task quality **without any assumption on the task heterogeneity**. We provide a summary table below.
>
> | **Aspect**                       | **Byzantine-robust agg.**               | **Anomaly detection**                             | **DP-FL**                     | **MOS + TPC (ours)**                                                                 |
> |----|------|-------|------|-----|
> | **Defense format**              | Distance / coordinate-based robust aggregation             | Feature / distance / clustering-based detector    | Clipping + Gaussian noise(**Server-side**)        | Output-based safety & downstream performance filters                        |
> | **Main signal**                 | **Large pairwise distances**, outlier coordinates          | **Statistical outliers** in feature / norm space  | **Norm-bounded updates**, then **indiscriminate noise** on all updates | **High DTR** on jailbreak prompts; **Low EM improvement**                             |
> | **Core assumptions**            | Accurate attacker count; **low data heterogeneity**        | Benign updates form a **coherent compact cluster**| DP tuned for privacy with **moderate noise**      | Only server queries responses; **no IID assumption**                                  |
> | **Why fails on CloudGhost**     | Heterogenous local objectives; DPT's limiting on excessive updates | Heterogenous local objectives; DPT's limiting on excessive updates | DPT makes updates **survive practical noise**; Not tailored for defense | CloudGhost keeps **low DTR** and **high ASR and EM**, thus bypassing MOS & TPC checks |
> | **Impact on training**          | **Notable EM drop** in high heterogeneity                  | **EM instability**, many false alarms             | Stronger noise => **severe EM degradation**        | No impact on benign training; limited overhead                                       |
> | **Effectiveness in LLM Geo-distributed and FL training** | **Failed**                                 | **Failed**                                       | **Impractical** as a robust defense               | **Effective and practical against naive attacks; vulnerable to CloudGhost**           |

---

> ### Author Response · Authors · 2025-11-20
> **Ans for Q2 and references**
>
> **Q2 – Evidence/justification for geo-distributed LLM training.**
> > *Evidence for geo-distributed LLM training?*
>
> ***Answer to Q2:***
> **INTELLECT-1** [8] is the first geo-distributed trained 10B LLM using a fully decentralized PRIME framework. **OpenDiLoCo** [9] achieved around 90–95% utilization while training across two continents and three countries.
>
> **NVIDIA’s NeMo and Megatron-Core** frameworks now officially support **multi-data-center LLM training** with approximately 96% scaling efficiency across long-haul inter-DC networks. The post details a real 340B-parameter training case, confirming the potential[10].
>
> In academia, the **DiLoCo** [11] proposes distributed low-communication training across poorly connected islands. **FusionLLM** [12] presents a practical system for training LLMs on geo-distributed GPUs with adaptive compression, while **Atlas** [13] explicitly targets multi-data-center training over wide-area networks. **Petals** [14] enables collaborative fine-tuning and inference over the Internet.
>
> | **System**        | **Setting / Architecture**                         | **Key Feature**                                      | **Evidence**                        |
> |--------------------------|----------------------------------------------------|-------------------------------------------------------------|--------------------------------------------------------------|
> | **Commercial LLMs**      | Centralized **internal GPU clusters**             | **High compute / energy / infra cost**                      | Motivates **geo-distributed training**       |
> | **INTELLECT-1** [8]      | Fully decentralized **PRIME** framework           | **10B LLM** trained in geo-distributed setting              | First **geo-distributed 10B** LLM                            |
> | **OpenDiLoCo** [9]       | Cross-continent  | DiLoCo replication with **90–95% utilization**              | Demonstrates **efficient WAN training**                      |
> | **NeMo / Megatron-Core** [10] | Multi–data-center, long-haul **inter-DC**   | ~**96% scaling efficiency**; **340B**-parameter case        | **Commercial-grade** multi-DC LLM training                   |
> | **DiLoCo** [11]           | Poorly connected **“islands”**                    | **Low-communication** distributed training                  | Algorithmic foundation          |
> | **FusionLLM** [12]        | **Geo-distributed GPUs**                          | **Adaptive compression**                   | Practical **system implementation**      |
> | **Atlas** [13]            | **Multi–data-center**, WAN                        | System optimized for **cross-DC LLM training**              | Explicitly targets **WAN-based** LLM training                |
> | **Petals** [14]           | Internet-scale **collaborative network**          | Collaborative **fine-tuning & inference**                   | Demonstrates **crowd / Internet** LLM collaboration          |
>
> ---
>
> > ***References***
> > [1] Shejwalkar, V., et al. (2021). *Manipulating the Byzantine: Optimizing Model Poisoning Attacks and Defenses for Federated Learning.* NDSS 2021.
> >
> > [2] Li, S., et al. (2024). *An experimental study of byzantine-robust aggregation schemes in federated learning.* IEEE Transactions on Big Data.
> >
> > [3] Huang, W., et al. (2024). *Self-Driven Entropy Aggregation for Byzantine-Robust Heterogeneous Federated Learning.* ICML 2024.
> >
> > [4] Blanchard, P., et al. (2017). *Machine Learning with Adversaries: Byzantine Tolerant Gradient Descent..* NeurIPS 2017.
> >
> > [5] Baruch, G., et al. (2019). *A Little Is Enough: Circumventing Defenses for Distributed Learning.* NeurIPS 2019.
> >
> > [6] Fang, M., et al. (2020). *Local Model Poisoning Attacks to Byzantine-Robust Federated Learning.* USENIX Security 2020.
> >
> > [7] Van der Vlist, F. (2024). *Big AI: Cloud infrastructure dependence and the scaling of AI training.* *International Journal of Automation and AI Systems.*
> >
> > [8] S. Jaghouar *et al.*, *INTELLECT-1 Technical Report*, arXiv:2412.01152, 2024.
> >
> > [9] S. Jaghouar *et al.*, *OpenDiLoCo: An Open-Source Framework for Globally Distributed Low-Communication Training*, arXiv:2407.07852, 2024.
> >
> > [10] NVIDIA (2025). *Turbocharge LLM Training Across Long-Haul Data Center Networks with NVIDIA NeMo Framework.* NVIDIA Developer Blog. https://developer.nvidia.com/blog/turbocharge-llm-training-across-long-haul-data-center-networks-with-nvidia-nemo-framework/
> >
> > [11] A. Douillard *et al.*, *DiLoCo: Distributed Low-Communication Training of Language Models*, arXiv:2311.08105, 2024.
> >
> > [12] Z. Tang et al., *FusionLLM: A Decentralized LLM Training System on Geo-distributed GPUs with Adaptive Compression*, arXiv:2410.12707, 2024.
> >
> > [13] Palak *et al.*, *Improving training time and GPU utilization in geo-distributed language model training*, arXiv:2411.14458, 2025.
> >
> > [14] A. Borzunov *et al.*, *Petals: Collaborative Inference and Fine-tuning of Large Models*, arXiv:2209.01188, 2023.

---

> ### Comment · Reviewer_weRd · 2025-11-27
>
> Thank you for the detailed response. The results against stronger defensive settings make the proposed vulnerabilities more plausible, and the practices of federated learning in LLM training make the scenario of this paper more practical. Therefore, I raise my confidence score to 4 and keep my positive rating score.

---

### Author Response · Authors · 2025-11-29
**Rebuttal Summary for AC: Reviewer Decisions and Discussion**

Dear AC and reviewers,

We sincerely thank all reviewers and the AC for their time, thoughtful, and constructive discussions. We deeply appreciate that **after discussion, we have addressed almost all concerns raised by the reviewers.**

#### Reviewers’ Decisions After the Discussion

1. **Reviewer weRd** maintained the positive score (6) and **increased the confidence score from 2 to 4** after the discussion.
2. **Reviewer HtvH** offered **exceptional praise for our work's quality and motivation**, describing the paper as **"exceptionally well-written and well-structured"** and noting that we do a **"very nice job of contextualizing the work and motivates future work in light of the contemporary AI landscape"**. And our experiment is **"neatly scoped and compelling"** and the evaluation is **"comprehensive and well-designed."** The reviewer kept the **original positive score (8) with confidence 4**.
1. **Reviewer vMwJ** acknowledged our comprehensive responses and **raised the score from 2 to 4**.

#### Our contributions are summarized below:

* **Pioneering Threat Analysis:** We are the first to consider jailbreak stealthiness in geo-distributed and federated LLM training, uncovering a critical, underexplored security risk in decentralized paradigms (Supported by **Reviewer weRd, HtvH, vMwJ**).
* **Novel Attack Framework:** We propose *CloudGhost*, which elegantly combines pseudo-contrastive alignment with Fisher-based regularization to balance stealth and effectiveness, successfully bypassing stringent server-side defenses (Supported by **Reviewer weRd, HtvH, vMwJ**).
* **Extensive Empirical Evaluation:** Our comprehensive evaluation across **diverse safety-aligned LLMs**, **varying attacker scales**, and **various trigger forms** demonstrates CloudGhost's robustness and generality (Supported by **Reviewer weRd, HtvH**).

#### During the discussion phase, we provided detailed clarifications and additional results, summarized here:

* **Clarified Novelty & Context:** We rigorously **clarified the distinction between our setting and traditional FL backdoors**. We provided detailed analyses justifying why prior attacks (e.g., *FedLLM-Attack*, *PEFT-as-an-Attack*, *Neurotoxin*) **lack stealth** against our proposed **MOS** and **TPC** defenses.
* **Robustness against Advanced Defenses:** We **visualized the data heterogeneity** in geo-distributed and federated LLM training. We extended our evaluation to **include state-of-the-art FL defenses**, including **SDEA** (Entropy-based), **DnC** (Spectral-based), **Clipped Clustering**, **Multi-Krum**, and **Differential Privacy**. Results confirm that CloudGhost successfully bypasses these defenses without degrading downstream performance.
* **Scalability:** **We validated that 10 clients is a practical setting for geo-distributed LLM training (see our response to Reviewer vMwJ's Weakness 4)**. We also conducted large-scale experiments with **100 clients**, confirming CloudGhost's effectiveness.
* **Reproducibility:** We provided an anonymized code repository and detailed hyperparameter tables to ensure reproducibility.

#### Regarding Scalability and Attacker Ratio issue raised by Reviewer vMwJ:
While Reviewer vMwJ noted that a 20–50% attacker ratio is high for general FL, we respectfully argue that this is an **inherent trade-off in FL security** between **Attack Effectiveness** and **Attacker Quantity**, rather than a **flaw specific to CloudGhost**. This attacker ratio is **also supported with top-tier literature** (See our latest response to reviewer vMwJ). Due to the massive dilution effect of global aggregation, malicious updates must either be aggressively boosted (sacrificing stealth and triggering TPC/Norm-Clipping) or be present in sufficient quantity.

**We are grateful that almost all of the raised concerns have been addressed**. We are pleased that **Reviewers weRd and HtvH have maintained their positive scores (with confidence both 4)**, and that **Reviewer vMwJ has raised their score to 4**. **We hope the AC can carefully consider our discussion regarding the scalability issue.**

We have incorporated all changes into the revised manuscript. We again thank all reviewers and the AC for their valuable time and effort.

Best regards and thanks,

Authors of Submission 12133

---

### Meta-Review · Area_Chair_WQQJ · 2026-01-07

**Summary:**

Most concerns of the reviewers were addressed, except those from reviewer vMwj on the ratio of attackers remains unsatisfyingly addressed.
Reviewers engaged with the authors, as a result, reviewer vMwj (though being among the least familiar with the subject) already stated willingness to raise the score from 2 to 4, while reviewer weRd increased the confidence score (while maintaining the overall score of 6), reviewer HtvH already offered a high score of 8 and did seem to be leaning to increase further.

I recommend acceptance and invite the authors to further explore scenarios with lower (and more realistic) rates of attackers in future work.

**Reviewer Concerns:**

most concerns of the reviewers were addressed, except those from reviewer vMwj on the ratio of attackers remains unsatisfyingly addressed.

**Reviewer Scores:**

reviewers engaged with the authors, as a result, reviewer vMwj (though being among the least familiar with the subject) already stated willingness to raise the score from 2 to 4, while reviewer weRd increased the confidence score (while maintaining the overall score of 6), reviewer HtvH already offered a high score of 8 and did seem to be leaning to increase further.

---

### Decision · Program_Chairs · 2026-01-26

Accept (Poster)